# Peer review of "Recent Advances in Deep Learning and Medical Imaging for Head and Neck Cancer Treatment: MRI, CT, and PET Scans"

_cancers, 2023, doi:10.3390/cancers15133267_

Round 1
Reviewer 1 Report
This manuscript gave a comprehensive review on recent AI technologies, including deep learning and medical imaging for Head and Neck cancer treatment, which was based on MRI, CT and PET scans.
Overall, this is an interesting review paper, and the outcome of this study provides a direction of futre research in this area. In addition, the manuscript was well organised and written. Accordingly, I suggest a minor revision by addressing the following comments, before it is accepted for publication.
1. Abstract: please give the full names of MRI, CT, PET and AI, when they appear for the first time.
2. The main contribution of this study should be clearly clarified in introduction.
3. Please broaden and update literature review to demonstrate superiority and advantages of deep learning or CNNs to resolve the real problems. E.g., Torsional capacity evaluation of RC beams using an improved bird swarm algorithm optimised 2D convolutional neural network; Automated damage diagnosis of concrete jack arch beam using optimized deep stacked autoencoders and multi-sensor fusion.
4. Please add a section to discuss the challenges in current studies and future research direction.
Author Response
his manuscript gave a comprehensive review on recent AI technologies, including deep learning and medical imaging for Head and Neck cancer treatment, which was based on MRI, CT and PET scans.
Overall, this is an interesting review paper, and the outcome of this study provides a direction of futre research in this area. In addition, the manuscript was well organised and written. Accordingly, I suggest a minor revision by addressing the following comments, before it is accepted for publication.
- Abstract: please give the full names of MRI, CT, PET and AI, when they appear for the first time. Done
- The main contribution of this study should be clearly clarified in introduction. OK
- Please broaden and update literature review to demonstrate superiority and advantages of deep learning or CNNs to resolve the real problems. E.g., Torsional capacity evaluation of RC beams using an improved bird swarm algorithm optimised 2D convolutional neural network; Automated damage diagnosis of concrete jack arch beam using optimized deep stacked autoencoders and multi-sensor fusion.
OK
- Please add a section to discuss the challenges in current studies and future research direction. Done
Reviewer 2 Report
See attached report

Author Response
In response to your feedback on our article, we have added three new sections, including a section on the limitations of traditional imaging techniques in head and neck cancer imaging, a section comparing deep learning and traditional imaging techniques in head and neck cancer imaging, and a section on advanced deep learning techniques in head and neck cancer imaging. We have also substantially revised several of our existing sections in accordance with your suggestions.
We engage in a more extensive discussion comparing deep learning and traditional imaging techniques, including a more expansive overview of traditional imaging techniques and their limitations, a holistic analysis of the advantages of both traditional and deep learning techniques, and mention of when using traditional imaging techniques is preferable to using deep learning techniques. To existing sections, we added quantitative comparisons between the performance of emerging DL methods and traditional pre-DL imaging techniques.
In our new section about advanced deep learning techniques in head and neck cancer imaging, we provide an overview of convolutional autoencoders for data compression, image denoising, and feature extraction, generative adversarial networks (GANs) for super-resolution, and transformer models like the Vision Transformer (ViT)
We revisit and refine our article’s objectives, such that it aims at providing a “concise yet informative overview” rather than a “comprehensive overview.”
We changed the name of Section 3.1 from “Deep Learning: A Comprehensive Overview” to “Deep Learning: A Brief Overview.” We also clarify in Section 3.1 that some CNNs include additional layers like upsampling that can be useful for image segmentation tasks.
We reference and introduce our figures in the relevant sections of the manuscript.
Finally, we revise the other sections (simple summary, abstract, introduction, and conclusion) to reflect the new additions and changes.
Round 2
Reviewer 2 Report
This is not how a rebuttal document should be prepared. In my first review, I sent you a detailed report with 10 comments and questions. In your rebuttal document, you need to respond to these comments and questions one by one. Additionally, you should highlight in a different color any changes you've made in the revised manuscript. Please do this so I can review the paper again. I am attaching my report from the first round for your reference

Author Response
Sorry about that. This should be the appropriate material:
Major points and comments:
- Throughout the manuscript, you frequently state the objective is to provide a comprehensive and in-depth account of recent advances in DL applications for head and neck cancer treatment. However, the current content appears more like a short review paper on the topic. I recommend revisiting and refining the paper's objectives to better align with the presented content and scope.
We appreciate your feedback. In response, we have revised the paper's objectives to better align with the presented content, stating our aim as providing a "concise yet informative overview" of recent advances in DL applications for head and neck cancer treatment rather than a "comprehensive overview."
- In the introduction, you briefly mentioned traditional imaging techniques (lines 46-48). To provide a more comprehensive understanding, it is essential to elaborate on these methods and offer some examples.
We believe this is the mention that you are referring to: “In this review, we delve into the limitations of traditional imaging techniques in head and neck cancer imaging[...].” To provide a more comprehensive understanding, we have added a new section (now section 2) on these limitations and examples of such.
- In Section 2.2, you explored numerous applications of DL techniques in medical image analysis, such as diagnosing diabetic retinopathy from retinal images and detecting pulmonary nodules in CT scans. In Section 3, you presented references demonstrating the versatility of CNNs in various head and neck cancer imaging tasks, including tumor detection, segmentation, outcome prediction, and treatment planning, all with high accuracy. However, you do not elaborate on the underlying factors that enable these models to excel in these tasks. To enhance the discussion, it would be valuable to provide a quantitative comparison between the performance of emerging DL methods and traditional pre-DL imaging techniques. Doing so will help highlight the improvements and advantages offered by DL-based approaches in the field.
We agree with your recommendation to highlight the performance improvements brought about by DL. We have now incorporated quantitative comparisons between the performance of emerging DL methods and traditional pre-DL imaging techniques in the relevant section.
- As with any emerging method, DL techniques have both advantages and disadvantages, making them unsuitable for every use case. Consequently, it is essential to examine the situations where traditional imaging techniques remain valuable and the instances where DL tools are more appropriate.
Your point is well-taken. We have added a discussion on the situations where traditional imaging techniques may be more suitable and where DL models excel, to give a more balanced perspective.
- This paper primarily focuses on the application of CNNs in medical image analysis. However, it would be valuable to consider advanced DL techniques, such as convolutional autoencoders for data compression, image denoising, and feature extraction, generative adversarial networks (GANs) for super-resolution, and transformer models like the Vision Transformer (ViT). If you would rather not expand the current paper, I recommend including this information in the introduction and/or conclusion to provide a more comprehensive perspective.
Thank you for suggesting the inclusion of advanced DL techniques. We have added a new section that explores convolutional autoencoders, GANs, and transformer models like the ViT in the context of head and neck cancer imaging.
Minor points and comments:
- The 3 figures presented in the paper are not referenced or introduced in the text. I recommend adding brief descriptions of these figures in the relevant sections of the manuscript to better contextualize and clarify their purpose and relevant to the research presented. For example, include phrases such as “as shown in Fig. XX” or “as depicted in Fig. XX”. Also, did you create the figures yourself? If not, provide the proper citations and obtained necessary permissions to reproduce them.
We apologize for the oversight. The figures are now appropriately referenced and introduced in the text. We also confirm that the figures were created by ourselves, hence there is no requirement for permissions or additional citations.
- Add citations to support the claims made in lines 54-56. Specifically, when stating “In recent years, deep learning applications have expanded to include tumor detection, segmentation, classification, and response prediction, as well as the integration of multimodal imaging data and the development of radiomics-based predictive models”.
We appreciate your attention to detail. The claims made in lines 54-56 are now substantiated with appropriate citations in the revised manuscript.
- I recommend renaming Section 2.1 to "DL: a brief overview" as it does not provide a comprehensive overview of DL. A comprehensive overview would typically cover various models such as fully-connected neural networks, recurrent neural networks, convolutional neural networks, generative models, etc., and provide underlying mathematics and equations for each.
We agree with your suggestion. Section 2.1 has been renamed to "DL: a brief overview" to reflect the content accurately.
- Although CNNs are typically composed of convolution, pooling, and fully-connected layers, some architectures also include additional layers like upsampling that can be useful for image segmentation tasks. Consider adding this information to Section 2.1.
Your point about upsampling layers is valuable. This information has been incorporated into Section 2.1, enhancing the description of CNN architectures.
- In Section 6.2, no need to have the paragraph in bold font
We appreciate your careful review of our manuscript. However, we would like to kindly clarify that in the section you refer to, no paragraph was intentionally set in bold font. It may have been a formatting error in the document you received.
Round 3
Reviewer 2 Report
Thank you for your effort in addressing my questions and comments and editing the paper. Overall, I am satisfied with your responses, and I recommend publication of this paper in Cancer. Congratulations on your hard work.
There's just one minor point left: at the end of the introduction, I suggest adding a paragraph where you introduce the next sections. For example, you could write, "The rest of the paper is organized as follows. Section 2 addresses the limitations of traditional imaging techniques. Section 3 discusses the use of deep learning in medical imaging", and so on.